# Dietary Contaminants and Their Effects on Zebrafish Embryos

**DOI:** 10.3390/toxics7030046

**Published:** 2019-09-07

**Authors:** Marc Tye, Mark A. Masino

**Affiliations:** Department of Neuroscience, University of Minnesota Twin-Cities Minneapolis, MN 55455, USA

**Keywords:** zebrafish diet, heavy metals, contaminant, toxin, development, behavior, persistent organic pollutant

## Abstract

Dietary contaminants are often an over-looked factor in the health of zebrafish. Typically, water is considered to be the source for most contaminants, especially within an aquatic environment. For this reason, source water for zebrafish recirculating systems is highly regulated and monitored daily. Most facilities use reverse osmosis or de-ionized water filtration systems to purify incoming water to ensure that contaminants, as well as pathogens, do not enter their zebrafish housing units. However, diets are rarely tested for contaminants and, in the case of manufactured zebrafish feeds, since the product is marketed for aquaculture or aquarium use it is assumed that the feed is acceptable for animals used for research. The following provides examples as to how contaminants could lead to negative effects on development and behavior of developing zebrafish.

## 1. Introduction

Water is generally thought of as the medium in which deleterious compounds enter a fish’s body. In fact, most toxicological studies expose fish to toxins dissolved in a water-based solution. Toxins in solution can easily be transported internally, not through the skin which is generally impermeable in adults, but via permeable membranes such as gill epithelia, narial mucosae, and oral mucosae. Fish embryos, such as zebrafish, readily absorb toxins due to the permeability of the chorion and vitelline membranes. Source water in recirculating systems for zebrafish housing units is highly regulated and monitored in order to control for the presence of toxins. Most zebrafish facilities use reverse osmosis or de-ionized water filtration systems to purify incoming water to ensure that contaminants, as well as pathogens, do not enter the housing units. Additionally, ultraviolet (UV) sterilization is used in recirculating systems to irradiate bacteria and viruses that may exist in the system before the water is returned to fish tanks. Therefore, considerable time, effort, and infrastructure is used to manage and monitor water quality within recirculating systems.

Deleterious compounds can also enter a fish’s body via consumption of food. Live and manufactured feeds used in the culture of zebrafish have been known to contain deleterious compounds due to natural or human sourced toxins [1,2]. Absorption of nutrients, including toxins, mostly occurs in the small intestine. Once in the body, several fish species, including zebrafish, have been shown to exhibit maternal transfer of toxic compounds to oocytes [3,4,5,6,7,8,9,10]. Thus, contaminants consumed by adult zebrafish can be passed on to their offspring.

Zebrafish diets consist of manufactured feeds and/or live organisms. Manufactured feeds are often formulated for and sold to the food–fish aquaculture and aquarium industries. Live feeds consist of primarily brine shrimp (*Artemia* sp.), rotifers (*Brachionus* sp.), and paramecia. Generally, zebrafish diets are not tested for contaminants. In the case of manufactured zebrafish feeds, since the product is marketed for aquaculture or aquarium use it is assumed that the feed is acceptable for animals used for research. This is a severe and potentially costly (money, time, loss of important lines etc.) oversight as the objectives of feeding food–fish, aquarium fish, and laboratory fish differ greatly. Live organisms are also susceptible to contamination. The culture method, source, storage protocol, and processing of live diets can vary greatly between vendors and each have the potential to introduce contaminants.

At present, quality control of zebrafish diets consists of an assumption that the manufacturer and vendor produce and sell a “quality” product. While many of these manufactured feeds are considered quality products by their specific industry (food–fish or aquarium fish), they may not meet the unique quality control needs of laboratory animals. Currently, the only way to analyze feed for dietary contaminants is for an individual zebrafish facility to send samples into a lab for custom analysis. The cost of analyzing each batch of feed for every contaminant is prohibitive for all but the largest and/or well-funded facilities. As a result, the zebrafish community lacks information as to which contaminants should be tested and what concentrations of said contaminants are acceptable. This is diverges from the rodent model where the National Institutes of Health National Toxicology Program (NTP) requires specific open formula diets for rodents used in reproductive and developmental toxicity studies [11]. Each batch of rodent diets must be tested for contaminants and may not exceed limits set by the NTP [11].

Contaminants have been detected in companion animal feeds, rodent chow, and fish feeds [2,12,13,14,15]. In most incidences, the contamination levels are well below the threshold that is considered a health risk for humans. However, it is unknown if the levels present affect research outcomes. In extreme cases, dietary contaminants could result in high mortality of zebrafish embryos. One reported incident occurred at the University of Minnesota and University of Utah zebrafish facilities in 2016. Abnormally high levels of chromium were found in brine shrimp cysts that were fed to adult zebrafish which resulted in major malformations, discoloration, and near 100% mortality of zebrafish embryos [2].

While dietary contamination incidences resulting in zebrafish mortalities are relatively rare, it is possible that incidents of dietary contamination resulting in sublethal effects may be common. Knowledge of these incidences is minimal as zebrafish feeds are not tested for contaminants, many physiological effects of contaminants are not known, and the acceptable limits of contaminants are not yet determined. The Guide for the Animal Care and Use of Animals acknowledges that dietary contaminants may be present in diets and could affect experimental results [16]. Thus, it is likely that the presence of contaminants in feed produce subtle sublethal effects. However, such a determination is not possible without a better fundamental understanding of the contaminant source, type and/or levels that are present. This article will highlight the contaminants that are likely to enter a zebrafish diet (Table 1) and have been shown to affect the offspring of those exposed.

## 2. Contaminants

### 2.1. Heavy Metals

Heavy metals are known to bioaccumulate in aquatic animals. Pelagic marine fishes are often processed into fishmeal and fish oil, which are two of the most common ingredients in fish feeds [1]. Therefore, heavy metals such as arsenic, lead, mercury and cadmium are potential contaminants in formulated fish feeds [1]. Brine shrimp are the most common zebrafish feed, many of which originate from The Great Salt Lake, which is known to accumulate heavy metals such as selenium, mercury, lead and arsenic. Contamination is so prevalent that the Utah Department of Environmental Quality has declared that there is a potential risk to human consumption of waterfowl in the area [32], which consume brine shrimp and other invertebrates that inhabit the lake [33,34]. In a recently reported study, brine shrimp cysts at the University of Minnesota and University of Utah zebrafish facilities were found to contain chromium levels more than 30 times greater than any other zebrafish diet tested [2]. Offspring from adults fed the contaminated diets exhibited orange coloration in the yolks, cardiac edema, misshapen yolk sacs, developmental delay, lack of swim bladder inflation, and high mortality [2].

Mercury, selenium, chromium, and cadmium are known to be maternally transferred to oocytes in fish [10,35,36,37]. Maternally transferred cadmium has been shown to alter gene expression, retard development, and increase incidences of pericardial edema in larval zebrafish [36,38]. Offspring of zebrafish that consumed methylmercury have been shown to exhibit hyperactivity at 7 and 16 dpf when compared to control [39]. Maternally transferred methylmercury resulted in altered behavior in young Atlantic croaker (*Micropogonias undulatas*) [40]. Numerous developmental and behavioral defects have been observed in zebrafish embryos and larvae that were exposed to heavy metals in solution [41,42,43,44,45,46,47,48]. Further research involving maternally transferred heavy metals will likely demonstrate similar results.

### 2.2. POPs

Persistent organic pollutants (POPs) encompass a large array of insecticides, herbicides, and industrial chemicals that persist in the environment long after their intended use. POPs can bioaccumulate in the food chain via long-term, low-level contamination or from short-term high-level contamination as a result of industrial accidents [27]. Bioaccumulation occurs in the lipids of animals and, thus, fish meal and fish oils, which are used in formulating fish feeds [25,28]. POPs have been detected in the feeds of multiple fish species including tilapia (*Oreochromis mosssambicus*) [13], gilthead sea bream (*Sparus aurata*) [49], rainbow trout (*Oncorhynchus mykiss*) [17], and salmon [26,29].

Polychlorinated biphenyls (PCBs) are a class of industrial chemicals that are regularly detected in fish oils [27]. PCBs at a concentration of 11.4 ppm was detected in eggs of adult rainbow trout (*Oncorhynchus mykiss*) that were fed a contaminated commercial diet [27]. An increase in embryo mortality and alteration of swimming behavior has been observed in offspring of adult zebrafish exposed to PCBs [5,50].

Tebuconazole is a common fungicide used in the production of grains throughout the world. Tebuconazole has been shown to be maternally transferred to zebrafish embryos resulting in decreased heart rate in developing larvae [51]. Another fungicide, azoxystrobin, was shown to alter mortality and development of zebrafish embryos when exposed to adult zebrafish [52].

Brominated flame retardants (BFRs) are a group of industrial chemicals that have been detected in wildlife and even humans. These contaminants have been detected in zebrafish embryos whose parents consumed a diet containing BFRs [53]. BFRs have been shown to increase malformations, alter gene expression and reduce survival rates of zebrafish embryos, when exposed in solution [54,55]. A decrease in hatching rate, inhibition of growth, inhibition of acetylcholinesterase activity and decrease locomotion was observed in the offspring of adult zebrafish exposed to polybrominated diphenyl ethers (PBDEs), another type of BFR [8,56].

### 2.3. Hormones

Phytoestrogens are found in many plant-based feedstuffs used for formulating fish feeds including soy, cottonseed, barley, rice, wheat, and oat [1,27]. As the name implies, phytoestrogens are estrogenic in nature and include isoflavones, lignans, and coumestans [1,27].

Estrogenic activity was detected in 17 commercial diets, of which included TetraMin, a common zebrafish feed [30]. Quesada-Garcia et al. (2012) detected estrogenic or thyroid activity in all 32 commercial diets tested in their study, though the identity of those diets were not disclosed [57]. Admittedly, there is little information regarding the amount of dietary phytoestrogen that is considered safe for fish consumption [1].

Synthetic hormones have been added to formulated fish feeds in order to produce monosex populations, increase growth, improve reproduction, or to sterilize fish [1,58,59]. Further, feeds marketed for aquarium fish have been known to contain synthetic hormones which can be used to enhance the color or induce spawning of some aquarium fish species. The United States Food and Drug Administration (FDA) does not allow the use of growth hormone, thyroid hormones, gonadotropin, or other steroids on fish destined for human consumption; thus these are not allowed in fish feeds [1]. However, many zebrafish diets originate from outside of the United States and regulations regarding synthetic hormones in fish feeds may differ by country.

Dietary hormones have been shown to increase vitellogenin concentrations in male goldfish (*Carassius auratus*) [60], Medaka (*Oryzias latipes*) [61], fathead minnow (*Pimephales promelas*) [62], and tilapia (*Oreochromis mosssambicus*) [63]. In addition to being a major source of the lipid and amino acid nutrients in developing larvae, vitellogenin and its derived yolk proteins are immune competent molecules which provide antibacterial and antioxidant roles in developing embryos [64,65,66]. Vitellogenin has been shown to neutralize the infectivity of infectious pancreatic necrosis virus in Atlantic Salmon (*Salmon salar*), cause lysis of bacteria, and enhance macrophage phagocytosis [67,68]. Phosvitin, a protein derived from vitellogenin, can function as a microbial agent in zebrafish embryos and is capable of killing microbes such as *Escherichia coli*, *Aeromonas hydrophila*, and *Staphylococcus aureus* [69]. The degree to which vitellogenin concentrations affect immunological studies utilizing zebrafish is not understood.

## 3. Synergistic or Antagonistic Effects

Dietary contaminants may also influence toxicity studies in synergistic or antagonistic ways. As an example, selenium, while toxic by itself, can act as an antioxidant. Selenium has been shown to protect zebrafish against oxidative stress caused by exposure to cadmium [70] and reduce methylmercury toxicity [71]. Maternal transfer of mercury was reduced when adult zebrafish were fed diets containing elevated levels of selenium [72]. Lead and BDE-209, a flame retardant, were found to have synergistic effects on thyroid hormone content and reactive oxygen species generation in zebrafish larvae [73,74].

These effects are not limited to heavy metals. For example, genistein, a phytoestrogen commonly found in plant feedstuffs, can suppress toxicity effects of polycyclic aromatic hydrocarbons [75]. In contrast, genistein can react synergistically with bisphenol A, a compound found in some plastics, to increase its toxic effects [76].

## 4. Current Dietary Regulations and Oversight

In the United States, the FDA has set action levels for some potential contaminants that may occur in pet food and food fed to animals destined for human consumption [77,78], though regulations differ between the two feed types and batch specific testing is not required. *The Guide for the Care and Use of Laboratory Animals* does not specify acceptable levels of contaminants in aquatic feeds, rather it specifies that animals should be fed uncontaminated diets [16].

Directive 2010/63/EU regulates the use of animals used for research in the European Union (EU) while the *Code of Practice for the Housing and Care of Animals Bred, Supplied or Used for Scientific Purposes* regulates specifically for the UK. Regardless, both state that animal feeds should be “non-contaminated”, though specific levels that constitute contamination are not given [18,19]. Directive 2002/32/EC regulates the amount of undesirable substances in animal feeds for the EU, which includes “…animals belonging to species normally fed and kept or consumed by man…” [20].

Though regulations on dietary contaminants do exist and meet the needs for animals entering the human food-chain, they may not be stringent enough for the needs of laboratory animals. Further research is needed to determine the sublethal effects that dietary contaminants have on zebrafish.

## 5. Conclusions

Dietary contaminants have been shown to be present in zebrafish diets and it is likely that their presence produces a range of subtle sublethal effects on zebrafish adults and offspring (Table 2). More information is needed regarding sublethal effects that dietary contaminants have on zebrafish. Before this can be done, dietary contaminants must first be identified and occurrence assessed.

Guidelines for undesirable substances in laboratory animal feeds is understandably vague. However, strict dietary guidelines should be established for specific research areas such as toxicology, developmental biology, immunology, and neuroscience. The National Toxicology Program has established such guidelines for rodents and rabbits and should move toward similar guidelines for zebrafish.

Though diet is a principal route for contaminants to enter laboratory animals, testing and reporting of dietary contaminants is essentially nonexistent within the zebrafish community. Granting organizations and industry journals must demand detailed reporting of diet regimes including feed type, quantity fed, nutrient content, and contaminant levels. These details go beyond what is specified in the ARRIVE (Animal Research: Reporting of In Vivo Experiments) guidelines, which is endorsed by many peer-reviewed journals. Reporting diet regimes by simply stating “fed per standard zebrafish protocol” or “fed brine shrimp nauplii” does not provide sufficient detail to reproduce the study and should not be acceptable in peer-review publications. A feasible way to accomplish the amount of detail that is needed would be to establish open formula diets and conduct batch testing of nutrient and contaminant content. An open formula diet is a feed where the concentrations of all ingredients are publicly available. Open formula diets are needed for zebrafish to control for extrinsic dietary factors and increase reproducibility of research results, particularly for studies monitoring gene expression, embryo/larval development, oxidative stress, and behavior.

Formulation of one standardized zebrafish diet is not considered feasible since research needs, and opinions as to what is best, vary greatly. However, the number of diets that are currently used by the zebrafish community is far too great to maintain consistency between facilities. Reducing the number of diets to even 10, would be a drastic improvement in standardization, though open formula diets would be the most appropriate.

The idea of establishing an open formula diet for zebrafish is not new; however, progress has stalled because many within the zebrafish community get “bogged-down” with the notion that an open formula diet should be the “best performing” diet, which is not the true objective. While an open formula diet should meet the nutritional requirements of zebrafish and thus provide adequate growth and reproductive success, it is primarily meant to facilitate standardization of care, not provide the best growth and/or reproduction. Therefore, in order to establish an open formula diet, zebrafish nutrient requirements must first be determined.

The National Institutes of Health (NIH) has multiple open formula diets for rodents. Many animal models that are less common than the zebrafish model, such as guinea pig, rabbit, dog, swine, and ungulates have at least one established open formula diet. The NIH should put forth resources towards defining the nutritional requirements of zebrafish and establishing an open formula diet, while the zebrafish community should debate less about what diet is “best” and more on how to establish a diet that is reproducible. 

## Figures and Tables

**Table 1 toxics-07-00046-t001:** Contaminants found in fish feeds and specific feedstuffs for the formulation of fish feeds.

Contaminant Group	Specific Contaminant	Found in Fish Feeds	Feedstuff Found In
Heavy Metals [2,17,18,19,20,21,22,23,24]		Yes	Fish meal, fish oil, poultry feather meal, plant meal, rice bran
	Arsenic [18,19,20,21]	Yes	Fish meal, fish oil, poultry feather meal, plant meal, rice bran
	Selenium [22]	Yes	
	Mercury [17,23,24]	Yes	Fish meal
	Lead [17]	Yes	
	Chromium [2]	Yes	
	Cadmium [17,20]	Yes	Plant meal
Persistent Organic Pollutants (POPs) [13,17,20,25,26]		Yes	Fish meal, fish oil, plant meal
	Polychlorinated biphenyls (PCBs) [20,23,25,26,27,28]	Yes	Fish meal, fish oil, plant meal
	Brominated Flame Retardants (BFRs) [20,23,28,29]	Yes	Fish meal, fish oil, plant meal
Hormones * [30]		Yes	Soybean oil cake, corn-gluten meal, cottonseed meal, wheat flour
	Phytoestrogens [1,31]	Yes	Soybean meal, lupin seed meal, cottonseed meal, alfalfa leaf meal

* Estrogenic activity detected.

**Table 2 toxics-07-00046-t002:** Summary of contaminants and their effects on zebrafish embryos and larvae due to exposure via parental diet, embryo exposure, or maternal exposure.

Contaminant Group	Specific Contaminant	Parental Diet	Embryonic Exposure	Maternal Exposure	Effects
Heavy Metals					
	Chromium [2,48]	69.6 mg/kg			Orange coloration, cardial edema, mishapen yolk sacs, developmental delay
			3–30 μM		Altered behavior, oxidative stress, immunotoxicity
	Cadmium [38,42,48]			35.6 μM	Altered gene expression, retarded development, pericardial edema
			100 μM		Reduced head size, reduced gene expression, impaired neurogenesis
			1–10 μM		Altered behavior, oxidative stress, immunotoxicity
	Mercury [39,41,44]	1 ppm			Increased hyperactivity
			25 ppb		Delayed response to stimulus
			6 μg/L		Impaired tail development
	Arsenic [43,47]		0.5–1.0 mg/L		Reduced survival, delayed hatching, retarded growth, malformation of spinal cord, abnormal cardiac function, altered cell proliferation
	Lead [45,46]		0.21–1.0 mg/L		Uninflated swim bladder, bent spine, yolk-sac edema, hyperactivity
			0.2 mM		Impaired neurogenesis
Persistent Organic Pollutants					
	Polychlorinated biphenyls (PCBs) [5,50]			1 umol/kg injection	Increased mortality
		515 ng/g			Increased activity, altered behavioral response
	Tebuconazole [51]			0.20 mg/L	Decreased survival, decreased hatching rate, developmental toxicity
	Azoxystrobin [52]			20 μg/L	Decreased survival, delayed development, altered gene expression
	Brominated Flame Retardants (BFRs) [8,54,55,56]			0.16 μg/L	Inhibition of acetylcholinesterase activity, downregulation of genes
			0.5 mg/L		Increased mortality, increased malformation
			1–10 μM		Delayed hatching, morphilogical abnormalities, increased mortality
				3 μg/L	Decreased hatching rate, inhibition of growth

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
