# Peer review of "Dietary Contaminants and Their Effects on Zebrafish Embryos"

_toxics, 2019, doi:10.3390/toxics7030046_

Round 1

Reviewer 1 Report

This short review is of great value to the zebrafish scientific community.

Almost all of the focus on contaminants that may affect zebrafish research has been on water-bourne contamination and feed-bourne contaminants have been majorly ignored. However, as this review clearly highlights, feeds may be an important source of contamination with toxic substances that affect research data in this model.

The manuscript is well structured and covers a broad set of literature. However, in order to be published some aspects need improvement and I would strongly suggest authors to consider the following:

2. should be a more general heading (e.g. Types of Contaminants) and then the several groups of contaminants described should come in sub-headings as 2.1-2.3 in the introduction line 34, UV filtration is not the correct term as it is not a filtration step per se. It should be UV sterilization in line 57, it should be further clarified what the authors mean by "quality control standards of laboratory fish" eithet by explaining which standards are these or provide references in lines 119-123, it is not clear if the concentration detected in trout is toxic and what was the concentration that gave rise to malformations in development and behavior in zebrafish in line 157, vitellogenin roles as immune competent molecules could be  further discussed regarding the impact on specific research, such as immunology or cancer in line 198 it is very limiting to just refer toxicology as an impacted research area in line 200-203 it should also be mentioned and explored the implementation of the ARRIVE guidelines already followed by some journals in lines 225-229 caution should be taken when referring only to specification rather than also standardization. Both are required in compromise and the zebrafish community should still continue to implement and foster the use of better feeds as there is already scientific evidence that some feeds currently being used perform worse than others regarding zebrafish health, growth and reproduction. 

Reviewer 2 Report

It is comparable limited topic for a review paper due to not enough scientific background on discussing dietary contamination of embryonic zebrafish. Therefore, I would suggest to broaden the topic to zebrafish, including adult and embryo, which might have more information. In addition, , author may need to discuss more about phenotype and maybe biological significancy that cause by each contaminant groups and specific contaminants. The most important information that papers should provide is the biological effects and mechanism. Moreover, please include the concentration ( or dosage), biological effects (or phenotype) of specific contaminant that mentioned in article, either with a form of table of in paragraph. Overall, the limited topic makes the limited information provided by this article and make it insufficient to be a review paper.

Round 2

Reviewer 2 Report

None.